# Using a 24 h Activity Recall (STAR-24) to Describe Activity in Adolescent Boys in New Zealand: Comparisons between a Sample Collected before, and a Sample Collected during the COVID-19 Lockdown

**DOI:** 10.3390/ijerph18158035

**Published:** 2021-07-29

**Authors:** Meredith C. Peddie, Tessa Scott, Jillian J. Haszard

**Affiliations:** Department of Human Nutrition, Dunedin 9016, New Zealand; tessa.scott@otago.ac.nz (T.S.); jill.haszard@otago.ac.nz (J.J.H.)

**Keywords:** 24 h activity recall, 24 h activity, device measurement, physical activity, sleep, screen time, COVID 19

## Abstract

Background: Tools that assess all three components of 24 h movement guidelines (sleep, physical activity, and screen use) are scarce. Our objective was to use a newly developed Screen Time and Activity Recall (STAR-24) to demonstrate how this tool could be used to illustrate differences in time-use across the day between two independent samples of male adolescents collected before and during the COVID-19 lockdown. Methods: Adolescent boys aged 15–18 years (*n* = 109) each completed the STAR-24 twice, *n* = 74 before lockdown and *n* = 35 during lockdown. Results: During lockdown more than 50% of the sample reported gaming between 10 a.m. and 12 noon, transport was not reported as an activity, and activities of daily living spiked at mealtimes. Gaming and screen time were more prevalent in weekends than weekdays, with the highest prevalence of weekday screen use (before lockdown) occurring between 8 and 9 p.m. Differences in estimates of moderate-to-vigorous physical activity prior to and during lockdown (mean difference (95% CI); 21 (−9 to 51) min) and sleep (0.5 (−0.2 to 1.2) h) were small. Total and recreational screen time were higher during lockdown (2 h (0.7 to 3.3 h) and 48 min (−36 to 132 min), respectively). Conclusions: The STAR-24 holds promise as a single tool that assesses compliance with 24 h movement guidelines. This tool also allows clear illustration of how adolescent boys are using their time (instead of only providing summary measures), providing richer data to inform public health initiatives.

## 1. Introduction

In line with many other countries including Canada [1] and Australia [2], the New Zealand physical activity guidelines for children and adolescents include recommendations on sleep and recreational screen time [3]. While there is a multitude of studies that assess physical activity, sleep or screen time in isolation, few studies exist that have assessed all three components together in the same sample. This may be because it is often difficult to combine the methods used to assess the individual components in a manner that then accurately represents time-use across a 24 h period in a contextually appropriate manner, that is not overly burdensome for both researchers and participants. While sleep and physical activity can be measured using accelerometers, accurate assessment of screen use during the same time period is rarely undertaken.

Recently there has been a renewed interest in use-of-time recalls to assess 24 h activity because they have the ability to provide rich contextual information [4], they are less subject to common bias that affects traditional questionnaires [5], and they result in better compliance than accelerometery [6]. However, most of the 24 h activity recalls currently in use require access to specialised computer programmes to administer them [4]. Therefore, the original aim of this study was to compare the results of a simple, paper-based, interviewer-administered 24 h activity recall with the results of accelerometery in a sample of male adolescents in New Zealand who were participating in the SuNDiAL (Survey of Nutrition, Dietary Assessment and Lifestyle) 2020 study.

However, in March 2020 when data collection for the SuNDiAL project had already begun, the COVID-19 global pandemic hit New Zealand. On 23 March it was announced that from 25 March New Zealand would enter into a 4 week lockdown, where everyone (except for essential workers) would be required to stay home, to exercise in their local neighbourhood, and to only make trips for food and medicine. Schools, sports facilities, gyms and playgrounds were all closed, and learning was conducted remotely [7]. What occurred in the following weeks was one of the strictest lockdowns in the world, with high adherence by the population [8]. While there is some evidence emerging that indicates decreases in physical activity and increases in screen time as a result of the global pandemic-related restrictions [9], most of these data are retrospective, based on self-reported changes [10,11], or small longitudinal studies that have not captured the entire 24 h day [12]. The lockdown in New Zealand meant that recruitment for SuNDiAL 2020 and the movement of accelerometers around the country to collect data was suspended. However, where possible data collection was continued remotely (over phone or video calls) with participants who had already enrolled in the study, providing an opportunity to describe the 24 h activity patterns of two groups—one before and one during lockdown. Therefore, the revised aim of this study was to describe the 24 h activity patterns measured via the Screen Time and Activity Recall (STAR-24) in samples of adolescent boys before and during New Zealand’s level 4 COVID-19 lockdown. A further aim was to show how this tool could be used to illustrate differences in 24 h time-use across the day for different populations in different environmental settings.

## 2. Materials and Methods

### 2.1. Study Design

The data used in this study were collected as part of the larger SuNDiAL (Survey of Nutrition, Dietary Assessment and Lifestyles) 2020 project. This study was a nationwide cross-sectional survey of male adolescents in New Zealand (NZ). The main aim of SuNDiAL 2020 was to describe the nutritional status, dietary habits, health status, attitudes and motivations for food choice, 24 h activity patterns and screen time of adolescent boys. Many of the methods mirror closely those used for the ‘sister’ study (SuNDiAL 2019) that focused on female adolescents and are described here [13]. The sample size required for the primary objective (to estimate mean nutrient intakes with a +/− 0.2SD 95% precision interval) was *n* = 100. To allow for drop-outs, incomplete data and small design effect from school clusters the aim was to recruit 150 adolescent boys. The study was approved by the University of Otago Human Ethics committee (Health): H20/004. Online informed consent was obtained from all participants and from parents/guardians for those under 16 years. The study is registered with the Australian New Zealand Clinical Trials Registry: ACTRN12620000185965.

To correspond with data collector ability and school terms, data collection was planned to be conducted in two phases, February to April 2020 and July to September 2020. However, recruitment was halted on the 23 March 2020 when it was announced the country was going into level 4 lockdown [7] as a result of the COVID-19 pandemic. Data collection continued for the recruited participants where it was possible to do so in an online setting.

### 2.2. Participants

Initially, all high schools who had male students enrolled with a total roll of greater than 400 that were in the predetermined data collection areas across New Zealand were emailed an invitation to participate. The five locations in the first phase (Otago, Christchurch, Wellington, Bay of Plenty, and Auckland) were chosen based on the living arrangements of the data collectors, who were second-year Master of Dietetics students from the University of Otago, Dunedin. Adolescent boys were then recruited from the schools through presentations to individual classes, year groups, or the whole school. Adolescents who were enrolled in one of the participating schools, were between 15 and 18 years of age, who self-identified as male and who could speak and understand English were eligible to participate.

### 2.3. Study Protocol

Most of the data collection was conducted at school during school hours. Prior to the in-school data collection appointments, participants provided consent and completed a questionnaire via an online REDCap survey. The questionnaire included basic demographic and health questions, as well as questions on dietary patterns, and attitudes and motivations towards food choice [13]. Each participant was then scheduled to participate in an in-school data collection visit that took approximately 60 min to complete. This in-school visit included anthropometric measures (weight and height), assessment of blood pressure, a 24 h diet recall, and the STAR-24 described here. A follow up data collection visit was performed the following week, over Zoom/Facetime/phone, to collect a second diet and activity recall. After the announcement of the lockdown, all 24 h recall data were collected remotely only (over phone or video calls) and the in-person collection of this data ceased.

### 2.4. Previous Day Screen Time and Activity

The STAR-24 was developed by the authors based on the Previous Day Physical Activity Recall [14], and the authors’ experiences of conducting 24 h dietary recalls [13]. Previous recalls of a similar nature have been shown to produce estimations of light and moderate intensity physical activity that are similar to estimates produced by accelerometry [4]. The STAR-24 was conducted using a multiple pass technique. In the first pass, participants were asked to identify the dominant activity for each 30 min time block from midnight to midnight of the day immediately preceding the day of the week data was being collected on. Data collectors categorised the reported activity into one of 13 possible categories that included sleep, schoolwork, activities of daily living, and physical activity at the time of administration (see Appendix A). When more than one activity occurred in a 30 min block, participants were encouraged to prioritise activities based on time, and then energy expenditure. In the second pass participants were asked to assign an intensity (sleep, very light intensity, light intensity, medium intensity or hard/vigorous intensity) to each activity they had identified in the first pass. Descriptions of these intensities were provided to the participants (see Appendix A). In the third pass, participants were asked to identify the posture they performed each activity in (lying, sitting, standing or stepping/moving). In the final pass participants were asked to identify which activities involved screen use as either none, phone, TV, computer/laptop or tablet/iPad. When more than one screen was used in the 30 min block participants were encouraged to identify the screen that was interacted with the most over that time period. Administration of each STAR-24 took ~15 min.

The same protocol was followed for a second recall that was carried out on a different day of the week so that, ideally, a weekend day and a weekday was captured for each participant. Weighted estimates of time spent in each category were calculated for each participant, where weekends contributed 2/7 and weekdays contributed 5/7. If participants only provided one day of data then estimates were not weighted.

### 2.5. Anthropometric Measurements

Measurements of weight (measured using one of Medisana PS420; Salter 9037 BK3R; Seca Alpha 770; or Soehnle Style Sense Comfort 400 scales) and height (measured using a Seca 213 or Wedderburn stadiometer) were taken in duplicate with the participant wearing light clothing and no footwear, and recorded to the nearest 0.1 kg or cm, respectively. Body mass index (BMI) was calculated and then converted to z-scores using the WHO child growth standards [15], and participants were classified as overweight if their BMI z-score was >1.

### 2.6. Statistical Analysis

All statistical analysis was carried out using Stata 16.1 (StataCorp, College Station, TX, USA). While the original aim was to validate the STAR-24 against accelerometry, the lockdown resulted in insufficient data to be able to do this.

To compare demographics between the samples before and during lockdown, unpaired, two-tailed t-tests were used for continuous variables and Fisher’s exact tests for categorical variables. To illustrate how the STAR-24 data can be used to describe the time use of the sample across 24 h, stacked area charts were used. These were split up to represent three different types of day: weekends, weekdays before lockdown, and weekdays during lockdown. Only one day from each participant was used in each plot—if they had two days available (e.g., if they had data for two weekdays before lockdown), then the first day was chosen. Some categories were further collapsed to more easily illustrate the data (these are described alongside each figure). Stacked area charts were generated for activities, intensity, posture, and screen use.

Estimated time in each activity, intensity, posture or using screens was described for the full sample using medians, 25th and 75th percentiles. As not everyone did each activity, this was also presented for those who did the activity, along with the number of participants and proportion of the sample.

To determine how lockdown might have influenced time spent asleep, in MVPA, and using screens, estimated durations from STAR-24 were used. As the data collected before lockdown were from a different sample than those collected during lockdown, linear regression models were used to estimate mean differences in time with adjustment for age, ethnicity, and deprivation. Residuals of models were plotted and assessed for heteroskedasticity and normality.

## 3. Results

### 3.1. Participants

All participants (*n* = 146) consented, completed online questionnaires and had anthropometrics collected prior to lockdown. Of those participants 109 completed at least one STAR-24, 74 of which were completed before lockdown, and 35 of which were completed during lockdown (Figure 1). The mean age (SD) of participants was 16.6 (0.7) years, 76% of the sample lived in areas of low to moderate deprivation and 67% were categorised as having a healthy BMI [15] (Table 1). The participants who completed the STAR-24 during lockdown were of a similar age (*p* = 0.894) but were more likely to: identify as Asian ethnicity (*p* < 0.001); live in an area of low deprivation (*p* = 0.005) and have a BMI z-score that categorised them as normal weight (*p* = 0.088), when compared to participants who completed the STAR-24 before lockdown.

### 3.2. 24 h Activity before and during Lockdown

Figure 2 presents data from STAR-24, with weekdays separated by lockdown status. Prior to lockdown, on weekdays more than 80% of the sample was awake by 8:00 a.m.; most of the sample reported schoolwork between 9:00 a.m. and 3:00 p.m. with screen use becoming more prevalent throughout the afternoon and evening. Approximately 80% of the sample reported being asleep by 10:00 p.m. During lockdown, transport disappeared as a reported activity, while gaming was markedly higher (more than 50% of the sample reported gaming between 10:00 a.m. and 12:00 noon). While there was only a small, non-statistically significantly higher amount of total sleep reported in those during lockdown (Table 2), sleep measured during lockdown appeared to happen later: 80% of the sample were asleep by midnight, with wakeup times closer to 10:00 a.m. The median times spent in each activity for the full sample, and for those who displayed the behaviour are presented in Appendix A.

Moderate to vigorous physical activity was not significantly lower during compared to before lockdown (Table 2). The pattern of accumulation of moderate-to-hard intensity activity did not differ dramatically across the sample before and during lockdown (Figure 3).

Total screen use was 2 h (95% CI 0.7 to 3.3 h) higher during lockdown, compared to before lockdown, although recreational screen time was not statistically significantly higher during lockdown (difference 0.8 h; 95% CI −0.6 to 2.2 h). However, Figure 2 and Figure 4 illustrate a marked difference in the pattern of screen use before and during lockdown: computers appeared to be more widely used throughout the middle of the day during lockdown, corresponding with the higher rates of gaming reported. TV use was more widespread during lockdown, particularly in the evening, while phone use seemed to be lower. Tablet use does not seem to make a large contribution to screen use in this group.

During waking hours sitting was the most predominant posture. Before lockdown, ~60% of the sample reported being seated at any one time. This appeared to hold true during lockdown, although there was a higher prevalence of sitting, and a lower prevalence of standing between 8 a.m. and 10 a.m. during lockdown compared to before lockdown (Figure 5).

## 4. Discussion

The results of this study illustrate that STAR-24 has the ability to describe 24 h activity patterns of adolescence boys with rich detail, and to illustrate differences in time use between the two samples collected prior to and during the level 4 COVID-19 lockdown in New Zealand. There did not appear to be a marked difference in total sleep between the two samples. However, the stacked area charts allowed for easy visualisation of the differences in when sleep was occurring. Most of the lockdown sample went to bed later and got up later in comparison to the pre-lockdown sample.

In contrast to other literature [9], MVPA of those in lockdown was not dramatically lower than the MVPA of those prior to lockdown. In fact, if anything the lockdown sample were doing slightly more MVPA than those who were assessed prior to lockdown. While parks and recreational facilities were closed, it is possible that messages to be physically active during the lockdown were more strongly expressed in New Zealand. For example, organisations encouraged the public to include physical activity in lockdown routines [16] and the state owned broadcaster (TVNZ) aired Les Mills’ virtual classes at 9 a.m. and 3 p.m. daily on two TV channels, with the classes also available on TVNZ on demand [17]. Interestingly, despite other measured components of the day differing in their timing (e.g., sleep), the timing of MVPA did not differ dramatically during lockdown, although timing of activity was variable with medium and hard intensity activities being reported at almost every time point during waking hours across the sample both prior to and during lockdown (i.e., at least one person reported during activity of moderate and hard intensity in almost every 30 min increment across the day).

Median leisure time screen time (screen time not for schoolwork) was higher than the recommended two hours a day for this age group both before (3.6 h) and during lockdown (5 h). Total screen time, however, was two hours greater in the sample under lockdown compared to those not in lockdown. Differences in total screen time appear to be driven mostly by an increase in gaming that was the predominant activity reported in the mid-morning during lockdown, with more than half the sample reporting gaming at around 10 am. This might suggest that boys were replacing schoolwork with gaming. However, at the beginning of lockdown (when most of these data were collected) many schools had teacher-only days to facilitate the move to online learning, and school holidays were brought forward [18], so it is likely that the boys did legitimately have less school work to complete over this time. Higher rates of screen time in adolescents have been associated with several poorer health outcomes including insulin resistance [19], higher BMI [20], and increased likelihood of metabolic syndrome [21]. Additionally, video games and computer use (as opposed to TV viewing) have also been associated with reporting of more severe depressive symptoms [22]. Clearly any health messages developed around ongoing or future lockdowns should consider targeting reductions in screen time.

The STAR-24 provided rich, contextual information on the time-use of samples of adolescent males before and during COVID-19 lockdown. Illustration of their time-use showed that gaming was a prevalent activity during lockdown. Other noticeable differences that occurred during lockdown include that no transport was reported during lockdown (compared to pre-lockdown), and patterns of activities of daily living changed markedly. While the reduction in transport is not unexpected given the lockdown restrictions [7], any reduction in active transport must have been replaced with other activity as overall moderate-to-hard intensity activity was not lower in those during lockdown. Interestingly, during lockdown activities of daily living seemed to spike around mealtimes, with ~50% of the sample reporting being involved in activities of daily living around lunch and dinner time. These peaks are also observed in weekends prior to lockdown, but the percentage of boys doing activities of daily living at any point during waking hours is higher, so these peaks are less exaggerated. It seems likely that these peaks represent the adolescent boys taking an active role in meal preparation or clean up but differentiating this from other activities of daily living was not possible. Given the paper-based administration of STAR-24, asking participants to differentiate activities of daily living into further sub-categories would be easily achieved if the focus of future research required it. The ability of STAR-24 to provide contextual information of this level is promising, particularly given that activities outside traditional MVPA may become the focus of future interventions in line with the new WHO recommendations that indicate that #everymovecounts [23].

Unfortunately, the lockdown meant that we were unable to collect enough accelerometer data to conduct any sort of validation of STAR-24. Ideally, in the future STAR-24 will be validated against a combination of data from wearable cameras and accelerometery collected in a manner that allows within participant comparisons of the same day. The use of both accelerometery and wearable cameras would allow for validation of not just estimates of sedentary time, physical activity and sleep, but also screen time and other contextual information that could not be validated by accelerometry alone. Clearly, however, this simple, easy-to-perform, paper-based activity recall holds promise as a means of assessing 24 h activity patterns, physical activity, posture and screen time at a population level. Moreover, it does so in a manner that is less burdensome to both participants and researchers, when compared to accelerometry, and more accurate when compared to other self-report measures [4].

### Limitations

The major limitation of the work presented here is that the original aim was not achievable given the COVID-19 lockdown, and the overall study was not designed to investigate the effects of a lockdown on 24 h time use. As a result, instead of comparing the activity of the same sample measured before and during lockdown using a validated assessment tool (which would have been the ideal), this study describes the 24 h activities of two samples of boys who differ in ways other than just their lockdown status using a tool that has yet to be formally validated. We have accounted for these limitations by adjusting for demographic differences in any comparisons. In doing so, we acknowledge that the differences presented here do not truly reflect changes to 24 h activity patterns as a result of the COVID-19 lockdown in New Zealand but rather differences between two similar samples in different lockdown situations.

STAR-24 itself does have some limitations that could be rectifiable by us or others during future validation studies. Upon reflection, and when compared to other 24 h activity recalls, 30 min increments of time are probably too big to represent some of the subtleties of movement accumulation, especially given that MVPA has been shown to mostly be accumulated in very short bouts in New Zealand female adolescents [24]. However, any smaller time increment would increase participant burden markedly and accuracy would depend on the ability of the participant to remember more detail about the previous day. While paper-based questionnaires require less resources to develop and administer, the presence of an interviewer may increase the likelihood of social desirability bias that may be lessened by self-administration. In addition, STAR-24 asked participants to identify the single screen that was most dominant in each 30 min time increment; however, the use of multiple screens at once has been reported in this age group. Therefore, future adaptions of STAR-24 could include the categorisations of secondary, and possible tertiary screens used in each time increment.

## 5. Conclusions

The COVID-19 lockdown in New Zealand likely impacted the 24 h activity of adolescent males, with more gaming during the day, and sleep timing—but not duration—shifting by about two hours. The STAR-24 holds promise to provide rich contextual information in a manner that is less burdensome to participants and researchers when compared to accelerometery and is likely more accurate than other self-report measures. It also has the ability to measure, with a single instrument, all the components of current physical activity guidelines for this age group, which, at least in New Zealand, contain messages about MVPA, sleep and screen time [25]. However, formal validation is still required.

## Figures and Tables

**Figure 1 ijerph-18-08035-f001:**
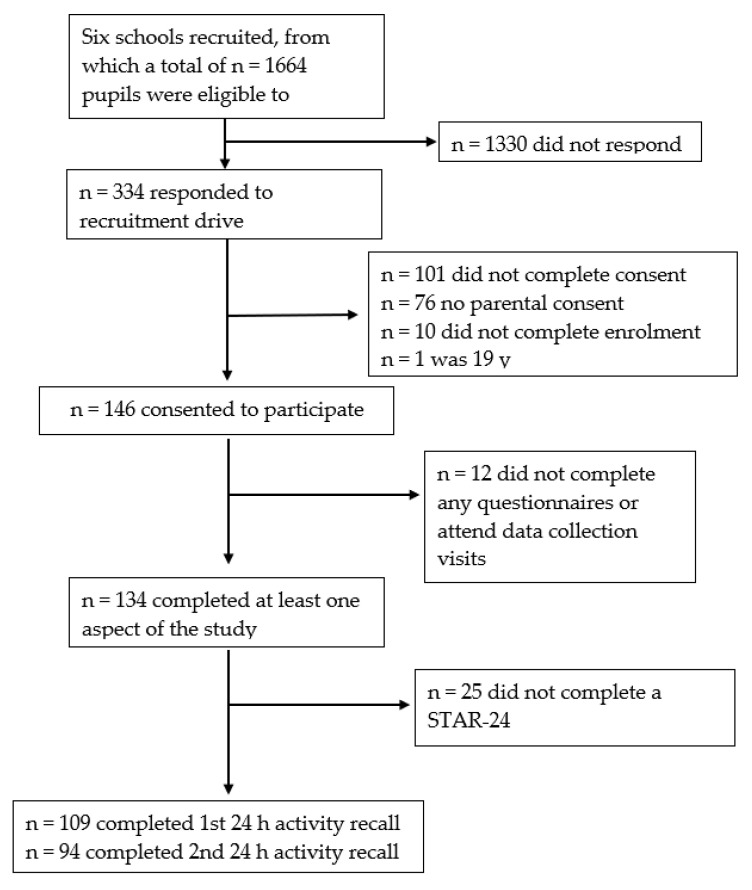
Participant flow chart.

**Figure 2 ijerph-18-08035-f002:**
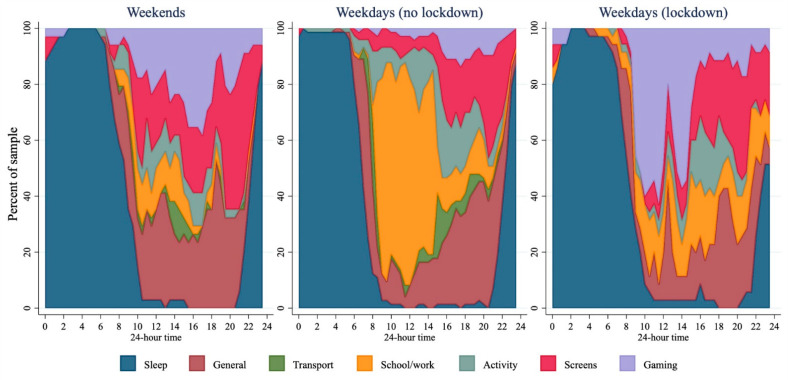
Activities across 24 h for weekends (*n* = 34), weekdays before lockdown (*n* = 73), and weekdays during lockdown (*n* = 35). ‘General’ included activities of daily living; ‘Transport’ included car/bus/train/e-scooter; ‘Activity’ included active transport, PE, sport, or other physical activity; ‘Screens’ included TV, computers, phones, not including gaming; and ‘Gaming’ included games played on PC, playstation, Xbox, Switch, mobile, etc.

**Figure 3 ijerph-18-08035-f003:**
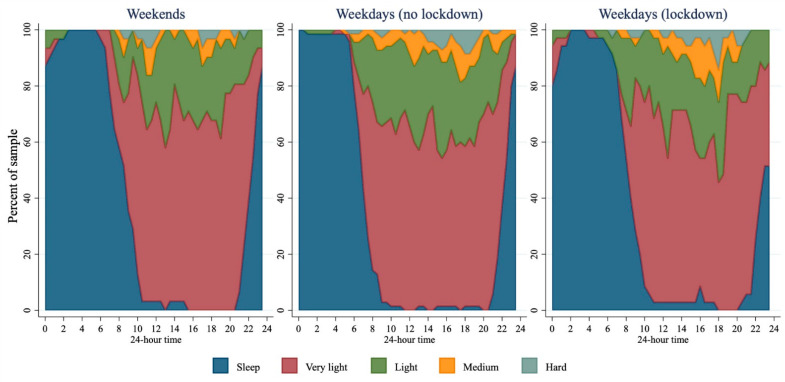
Intensity of activity across 24 h for weekends (*n* = 31), weekdays before lockdown (*n* = 72), and weekdays during lockdown (*n* = 35). Very light intensity—activities that involve very little or no movement, breathing rate is slow; Light intensity—activities that involved some movement, but do not elevate breathing rate; Medium intensity—moving quickly/briskly, breathing rate is increased but you can still talk; Hard/vigorous intensity—moving very quickly, breathing so hard you cannot talk at the same time.

**Figure 4 ijerph-18-08035-f004:**
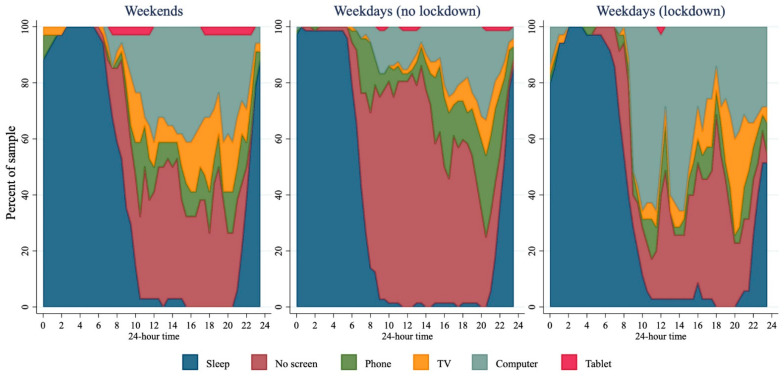
Screen use across 24 h for weekends (*n* = 34), weekdays before lockdown (*n* = 72), and weekdays during lockdown (*n* = 35). If multiple screens in use then the one they were interacting with most was prioritised.

**Figure 5 ijerph-18-08035-f005:**
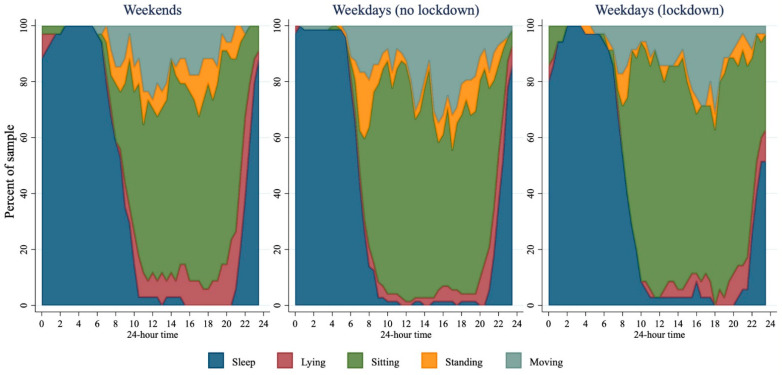
Posture across 24 h for weekends (*n* = 34), weekdays before lockdown (*n* = 72), and weekdays during lockdown (*n* = 35).

**Table 1 ijerph-18-08035-t001:** Characteristics of the samples (*n* = 109).

Characteristic	All	Completed STAR-24 before Lockdown	Completed STAR-24 during Lockdown
N	109	74	35
Age, mean (SD) years	16.6 (0.7)	16.6 (0.7)	16.6 (0.8)
Ethnicity, *n* (%)			
	NZEO ^a^	68 (61.8)	55 (73.3)
	Māori	10 (9.1)	10 (13.3)
	Asian	32 (29.1)	10 (13.3)
Household area deprivation ^b^, *n* (%)			
	Low	36 (34.6)	20 (28.6)
	Medium	43 (41.4)	27 (38.6)
	High	25 (24.0)	23 (32.9)
BMI z-score ^c^, mean (SD)	0.38 (1.15)	0.50 (1.15)	0.04 (1.09)
Weight status ^c^, *n* (%)			
	Normal weight	64 (67.4)	44 (62.9)
	Overweight	26 (27.4)	22 (31.4)
	Obese	5 (5.3)	4 (5.7)

^a^ NZEO—New Zealand European and Others, this includes those who chose not to specify (*n* = 4). ^b^ Household level deprivation measured using the NZDep2018 index split into deciles: Low (1–3); Medium (4–7); High (8–10). N = 4 participants were missing NZDep data. ^c^ BMI z-scores calculated using WHO growth charts, with overweight > 1 z-score and obese > 2 z-scores. N = 14 participants did not have BMI z-score data.

**Table 2 ijerph-18-08035-t002:** 24-h time-use factors by lockdown (*n* = 109).

	before Lockdown (*n* = 74)	during Lockdown (*n* = 35)	
	Median (25th, 75th percentiles)	Median (25th, 75th percentiles)	Mean difference (95% CI) ^a^
Sleep, hours	9.0 (8.0, 10.0)	9.8 (8.8, 10.3)	0.5 (−0.2, 1.2)
MVPA, minutes	45 (0, 90)	60 (30, 105)	21 (−9, 51)
Screen time (total), hours	6.4 (4.8, 9.0)	9.7 (8.0, 11.2)	2.0 (0.7, 3.3)
Screen time (outside of school work), hours	3.6 (1.8, 5.5)	5.0 (2.5, 8.0)	0.8 (−0.6, 2.2)

^a^ Mean difference (95% CI) estimated using regression models adjusted for age, ethnicity, and deprivation. N = 5 missing demographic data.

## Data Availability

The datasets generated for this study are available upon reasonable request to the corresponding author.

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
