# Peer review of "Using a 24 h Activity Recall (STAR-24) to Describe Activity in Adolescent Boys in New Zealand: Comparisons between a Sample Collected before, and a Sample Collected during the COVID-19 Lockdown"

_ijerph, 2021, doi:10.3390/ijerph18158035_

Round 1

Reviewer 1 Report

In section 2.4 Did the Data collectors categorize the data during administration or did it require multiple sessions.  Do the authors have any information on how long the test took to administer?  This would be helpful to know for possible implementation in other cohorts.  If this is specified in previous paper it would be useful to include in this one as well.

Table 2

Would have liked to see Before lockdown for N=35 that took second 24-h recall

I think it would be helpful for Table 2 or a supplement of table 2 to prove that there were no significant differences between the N=35 lockdown sample at baseline compared to the N=74 before lockdown.  Authors did attempt to account for differences by looking at demographic information but it’s also important to prove that time-use factors were not significantly different at baseline.

4.1 Limitations, just a comment, but I don’t think that the 30 min increment could be fixed using a smaller time increment.   While vigorous activity may occur in bursts, the correlation between the recall of a high activity event and total movement accumulation may be the same.   The original design of this study using accelerometry data would have done well at elucidating that relationship.

The study was interesting and would have provided important backing to the STAR-24 assessment had accelerometry been available. 

Author Response

Reviewer comment 1: In section 2.4 Did the Data collectors categorize the data during administration or did it require multiple sessions.

Response: The data was categorised at the time of administration.  A statement indicating this has now been added to the methods at line 122

Reviewer comment 2: Do the authors have any information on how long the test took to administer?  This would be helpful to know for possible implementation in other cohorts.  If this is specified in previous paper it would be useful to include in this one as well.

Response: The STAR-24 took approximately 15 min to administer.  A statement indicating this has been added to the methods at line 134.

Reviewer comment 3: Table 2. Would have liked to see Before lockdown for N=35 that took second 24-h recall I think it would be helpful for Table 2 or a supplement of table 2 to prove that there were no significant differences between the N=35 lockdown sample at baseline compared to the N=74 before lockdown.  Authors did attempt to account for differences by looking at demographic information but it’s also important to prove that time-use factors were not significantly different at baseline

Response: We agree it would have been good to have before data from this sample.  However, this group of participants only completed recalls during lockdown. There was no baseline before lockdown data to compare to.

Reviewer comment 4: 1 Limitations, just a comment, but I don’t think that the 30 min increment could be fixed using a smaller time increment. While vigorous activity may occur in bursts, the correlation between the recall of a high activity event and total movement accumulation may be the same. The original design of this study using accelerometery data would have done well at elucidating that relationship. The study was interesting and would have provided important backing to the STAR-24 assessment had accelerometery been available. 

Response:  Thank you for these comments.  We agree that accelerometery data would have been nice to have.  We hope to be able to do some more formal validation work with accelerometers and wearable cameras sometime in the future, and may very well investigate the impact of differing the recall increments.

Reviewer 2 Report

I wanted more detail on the implications of the amount of screen time spent by these boys. I found the paper top heavy on statistical analysis and light on the physical / emotional impact of lock down. 

Author Response

Reviewer comment 1: I wanted more detail on the implications of the amount of screen time spent by these boys. I found the paper top heavy on statistical analysis and light on the physical/emotional impact of lockdown.

Response: Thank you for this suggestion. The following has been added to the discussion at time 278

“Higher rates of screen time in adolescents have been associated with several poorer health outcomes including insulin resistance (19) , higher BMI(20), increased likelihood of metabolic syndrome (21).  Additionally, video games and computer use (as oppose to TV viewing) have also been associated with reporting of more severe depressive symptoms (22).  Clearly any health messages developed around ongoing, or future lockdowns should consider targeting reductions in screen time.”

Reviewer 3 Report

Thank you for allowing me to review this work. The manuscript is about the value of utilizing a 24-h activity recall scale to describe activity in adolescent boys during sleep, physical activity, and screen use. With the particular scale, the activity during the COVID-19 lockdown period has been monitored in relation to activity prior to the lockdown period.

One drawback is that the scale actually compared 2 different groups of adolescent boys to establish a difference in the domains this questionnaire monitors. In my opinion this should be reflected in the title of the study as well, otherwise the title is misleading. It is evident why this methodology was pursued, as any new recruit would have had to report a 24-recall of previous day activity level near their recruitment. However, given the inherent difficulties in making a direct comparison between the 2 groups, which were of unequal in size, ethnicity, areas of living and BMI, this methodology may lead to circumstantial evidence. Another limitation is that recall of the previous 24-hours activity may not provide an accurate representation of a wider activity timeframe.

Also, no validation against another activity measure or accelerometry was performed.

On the other hand, a relative advantage of the scale is that physical activity is assessed in multiple domains (sleep, physical activity and screen use).

Finally, there is no clinical message to this study. The accuracy of reporting also could have been checked with test-retest reliability

This description included in lines 62-63 may be somewhat misleading: unique opportunity to describe the 24-h activity patterns of two groups – one before and one during lockdown. Please revise.

The criteria upon which data collection was initially set to be performed across 2 periods were not clear (lines 84-85): ‘Originally data collection was planned to be conducted in two phases, February to April 2020 and July to September 2020.’

In reference 14 page numbers are missing.

Author Response

Reviewer comment 1: One drawback is that the scale actually compared 2 different groups of adolescent boys to establish a difference in the domains this questionnaire monitors. In my opinion this should be reflected in the title of the study as well, otherwise the title is misleading.

Response: Thank you for this suggestion.  The title of the manuscript has now been amended to: Using a 24-h activity recall (STAR-24) to describe activity in adolescent boys in New Zealand.  Comparisons between a sample collected before, and a sample collected during the COVID-19 lockdown.

Reviewer comment 2: It is evident why this methodology was pursued, as any new recruit would have had to report a 24-recall of previous day activity level near their recruitment. However, given the inherent difficulties in making a direct comparison between the 2 groups, which were of unequal in size, ethnicity, areas of living and BMI, this methodology may lead to circumstantial evidence.

Response: We agree that there are limitations involved in making comparisons between the two groups.  However, the comparisons between the groups have been adjusted to the obvious differences in ethnicity and deprivation between the two samples.  We clearly acknowledge these limitations in section 4.1

Reviewer comment 3: Another limitation is that recall of the previous 24-hours activity may not provide an accurate representation of a wider activity timeframe.

Response:  We agree that a 24-h recall will never adequately describe the activity of an individual.  However, in a similar manner to the way 24-h dietary recalls are used to describe the dietary intakes of groups in studies such as NHANES, we believe that multiple single-day recalls on different individuals can provide a reasonable approximation of the physical activity levels of the study population.  Particularly when recalls are performed across all days of the week (as was the case in the current study) to account for daily variability in activity.

Reviewer comment 4: Also, no validation against another activity measure or accelerometry was performed.

Response: We agree that accelerometery data would have been nice to have, but unfortunately this wasn’t possible given the level of COVID-19 restrictions that were in place in New Zealand.  We hope to be able to do some more formal validation work with accelerometers and wearable cameras in the future.

Reviewer comment 5: On the other hand, a relative advantage of the scale is that physical activity is assessed in multiple domains (sleep, physical activity and screen use).

Response: We agree that the ability of STAR-24 to assess multiple domains is a real strength of this instrument.

Reviewer comment 6: Finally, there is no clinical message to this study.

Response: in response to one of the other reviewer’s comments we have included more discussion (at line 280) about the possible negative impact of high amounts of screen time during lockdown and highlighted that any health messages that are designed as a result of future or ongoing lockdowns should target reductions in screen time.

Reviewer comment 7: The accuracy of reporting also could have been checked with test-retest reliability.

Response: Test-retest assessments are challenging for measures of short-term, actual behaviours (as opposed to long-term, usual behaviours), which may differ markedly from day to day. We hope to assess the reliability of the STAR-24 in future studies using both wearable cameras and accelerometers.

Reviewer comment 8: This description included in lines 62-63 may be somewhat misleading: ‘unique opportunity to describe the 24-h activity patterns of two groups – one before and one during lockdown’. Please revise.

Response:  The word unique has been removed from this statement.

Reviewer comment 9: The criteria upon which data collection was initially set to be performed across 2 periods were not clear (lines 84-85): ‘Originally data collection was planned to be conducted in two phases, February to April 2020 and July to September 2020.’

Response; Apologies for this oversight.  This statement now reads:

“To correspond with data collector ability and school terms data collection was planned to be conducted in two phases, February to April 2020 and July to September 2020.”

Reviewer comment 10: In reference 14 page numbers are missing.

Response: Thank you for drawing this to our attention.  Page numbers have been added to this reference.

Round 2

Reviewer 3 Report

This manuscript is now improved in 2 basic aspects: Title matching the methodology followed and some clinical implications of the findings. THis is fine by me now to proceed.